# Vaccination card loss and associated factors in Ethiopia: A multilevel analysis using Ethiopia Mini Demographic and Health Survey 2019 data

Zerihun Kura Edossa[1], Belay Erchafo Lubago[2]*, Minale Fekadie Baye[3], Rediet Kidane Alemu[4], Abebe Abera Tesema[5], Fira Abamecha[6], Yibeltal Siraneh[7], Dessalegn Tamiru[4], Negalign Berhanu Bayou[7], Gurmesa Tura Debelew[8]

1 Department of Epidemiology, Public Health Faculty, Institute of Health, Jimma University, Jimma, Ethiopia, 2 Department of Public Health, College of Medicine and Health Sciences, Wachemo University, Hossana, Ethiopia, 3 Department of Biochemistry, School of Biomedical Sciences, Jimma University, Jimma, Ethiopia, 4 Department of Human Nutrition and Dietetics, Public Health Faculty, Institute of Health, Jimma University, Jimma, Ethiopia, 5 School of Nursing, Faculty of Health Sciences, Jimma University, Jimma, Ethiopia, 6 Department of Health, Behavior and Society, Public Health Faculty, Institute of Health, Jimma University, Jimma, Ethiopia, 7 Department of Health Policy and Management, Public Health Faculty, Institute of Health, Jimma University, Jimma, Ethiopia, 8 Department of Reproductive Health, Public Health Faculty, Institute of Health, Jimma University, Jimma, Ethiopia

* erchafobelay@gmail.com

## Abstract

### Background

The loss of vaccination cards is a momentous public health challenge in the prevention of vaccine-preventable diseases in most developing countries. There is a paucity of studies on the magnitude of vaccination card losses and associated factors in Ethiopia. Therefore, the aim of the current study was to assess the level of vaccination card loss and associated factors in Ethiopia.

### Methods

Data were extracted from the 2019 Ethiopia Mini Demographic and Health Survey, a nationally representative household survey of women aged 15–49 years with children aged 0–35 months. Data from a total of 3208 mother-child pairs was extracted for the study. A multilevel logistic regression model with random effect analysis techniques was used to identify individual and regional-level determinants of vaccination card loss. We checked the model's fitness by using Akaike Information Criteria and Bayesian Information Criteria. Odds ratios with a 95% Confidence intervals were used to declare statistical significance.

### Results

A total of 3208 mother-child pairs from nine regions and two city administrations were included in the analysis. The result revealed that 1933 (60.26%) mother-child pairs did not have vaccination cards during home visits. illiterate mother 2.239 (95% CI: 1.297, 3.864),

**Data Availability Statement:** The original data set of 2019 EMDHS is available from EDHs program

on request using the following Link (https://dhsprogram.com/publications/index.cfm).

**Funding:** The author(s) received no specific funding for this work.

**Competing interests:** The authors have declared that no competing interests exist.

lowest wealth index category 2.089 (95% CI: 1.432, 3.048), ANC non-user 2.047 (95% CI: 1.605, 2.609), children living with their caretaker 6.749 (95% CI: 1.425, 13.654), having no access to television 1.384 (95% CI: 1.150, 1.664), longer birth interval 1.325 (95% CI: 1.027,1.710), giving birth at home and private health facilities 1.985 (95% CI: 1.579, 2.497), 1.696 (1.086, 2.648), contraceptive non-users 1.295 (95% CI: (1.042,1.609) and children aged 12–23 months and 24–35 months 1.577 (95% CI: 1.252, 1.985) and 2.282 (1.803, 2.889) were associated with vaccination card loss.

## Conclusions

Overall vaccination card loss among mothers of 0–35-month-old children is considerably high. To enhance maternal awareness regarding the significance of vaccination cards, promote antenatal care and public health facility delivery, and future researchers have to explore how to enhance vaccination card retention in Ethiopia.

## Introduction

Vaccination is one of the most significant and cost-effective public health interventions worldwide saving millions of people from infectious diseases death and disability and averting more than 12 million child deaths each year [1–3]. It enables the eradication of smallpox and lowers the incidence of polio by 99% and neonatal tetanus by 94% [4]. Immunization prevents about 750,000 children from disability every year by reducing the incidence of vaccine-preventable diseases(VPDs) [5, 6]. Vaccination also protects the unvaccinated and the community by conferring herd immunity [7] and reducing the spread of disease within a population [8]. Despite the effort to improve vaccination services, its coverage is very low in many parts of the African countries including Ethiopia [9–11].

Increased vaccination card retention is one of the key measures to evaluate the correct coverage rates and inform health policymakers on how to improve uptake [12]. The issuing and retention of vaccination cards are important indicators of the seriousness of the effort by the health system and the level of awareness among families. Low card retention and utilization have been shown by different studies in developing countries, including Ethiopia. Retention of vaccination cards is considered a proxy indicator to improve families' engagement in routine vaccination [13].

Families throughout the world possess vaccination cards, which at their base contain records of the dates their child was vaccinated with what vaccine, vaccine dose, lot number, ages, vaccination schedule, and sometimes detailed information and informed consent documents for various available vaccines [14]. Healthcare providers often give these documents to parents when they bring their child in for the first vaccination after birth [12, 14]. It also indicates to authorities, daycare providers, schools, doctors, and immigration authorities, to evaluate compliance, coverage, and practice in various settings, that a child has received certain vaccines, including ones mandatory for certain occasion [14–16]. Hence, accurate and legible vaccination records and cards serve as a comprehensive account of immunization services provided to an individual and should be part of an individual's and household's permanent vaccination cards [14].

However, a significant number of mothers either lost their child's vaccination cards or kept them so dirty that it is difficult to read what is written on the cards. This made it difficult for

both the mother and the medical personnel to ascertain what vaccines and dosages had been given [17].

Loss of a child vaccination card is likely to result in missed vaccination, repeated vaccination, and loss of important information about the health of the child, which will put the child's health and well-being at risk [17]. Low vaccination card retention is one of the important barriers that needs to be addressed for effective immunization programs across sub-Saharan Africa, including Ethiopia [7]. In Ethiopia, even if vaccination card retention is crucial in immunization service delivery to trace which vaccines were given previously and to know the appropriate dose, high vaccination card losses are still reported in the country. Vaccination cards are critical tools in ensuring that children receive all recommended vaccinations according to schedule. In Ethiopia, about 62% of children aged 0–35 months lost their vaccination cards during household visits in mini-EDHS-2019. Due to the lack of vaccination cards, mothers and care providers don't know which vaccine the children have taken and which doze. This in turn leads to the wastage of immunization resources and creates a favorable environment for outbreaks due to high defaulters [18, 19]. Previous studies in many countries, including Ethiopia, address different dimensions regarding vaccination utilization, prevalence, defaulters, and determinant factors. However, little emphasis was given to vaccination card loss, which is very critical to the utilization of vaccination services, defaulter tracing, preventing outbreaks, and maximizing the efficiency of immunization resources. In the Ethiopian context, currently, having a vaccination card is necessary to receive a birth certificate and for school enrollment. Hence, this study attempted to assess the level of vaccination card loss and its predictors in Ethiopia for effective immunization programs based on the data from the mini-Ethiopia Demographic and Health Survey (EDHS) 2019 data.

## Methods and materials

### Study area and study period

The data used in this study were extracted from the 2019 EMDHS. The survey was conducted by the Ethiopian Public Health Institute (EPHI) in collaboration with the Central Statistical Agency (CSA) and the Federal Ministry of Health (FMoH), with technical assistance from the International Classification of Functioning Disability and Health(ICF) and financial as well as technical support from development partners. The survey was conducted from March 21, 2019 to June 28, 2019, based on a nationally representative sample that provided estimates at the national and regional levels and for urban and rural areas. It used a survey design to collect data on respondents' background characteristics, fertility determinants, marriage, awareness and use of family planning methods, child feeding practices, nutritional status of children, childhood mortality, and height and weight of children aged 0–59 months. At the time of the survey, Ethiopia is a large country divided into nine regions and two city administrations.

### Sampling and data collection

A two-stage sampling technique was used to select representative samples of independent enumeration areas (EAs) in each stratum. Implicit stratification and proportional allocation were achieved at each of the lower administrative levels by sorting the sampling frame within each sampling stratum before sample selection, according to administrative units in different levels, and by using a probability proportional to size selection at the first stage of sampling. To ensure that survey precision was comparable across regions, sample allocation, was done through an equal allocation wherein 305 EAs were selected from regions and city administrations.

In the first stage, a total of 305 EAs (93 in urban areas and 212 in rural areas) were selected with probability proportional to EA size based on the 2019 Ethiopian population and housing census (EPHC) frame and with independent selection in each sampling stratum. A household listing operation was carried out in all selected EAs from January through April 2019. The resulting lists of households served as a sampling frame for the selection of households in the second stage. Some of the selected EAs for the 2019 EMDHS were large, with more than 300 households. To minimize the task of household listing, each large EA selected for the 2019 EMDHS was segmented. Only one segment was selected for the survey, with probability proportional to segment size. Household listing was conducted only in the selected segment; that is, a 2019 EMDHS cluster is either an EA or a segment of an EA.

In the second stage of selection, a fixed number of 30 households per cluster were selected with an equal probability of systematic selection from the newly created household listing. All women aged 15–49 who were either permanent residents of the selected households or visitors who slept in the household the night before the survey were eligible to be interviewed. In all selected households, women aged 15–49 were interviewed using the Woman's Questionnaire [20]. For this study a total of 3208 mothers whose children are alive and aged 0–35 months were included in the analysis.

## Statistical analysis

For the whole analysis STATA version 17 software package was used. Prior to the commencement of the analysis data cleaning, labeling, coding, and re-coding were done for all selected variables. Frequency and percentage were used to report categorical variables, while mean followed by standard deviation was used to report continuous explanatory variables. The model selection process was fitted including standard logistic regression models ignoring the cluster information. These models were fitted for comparison purposes; otherwise, they are completely not suitable for the data in this study. The fit statistics indicate a substantial improvement in the multilevel regression compared to the logistic regression models, implying the multilevel models were preferred over the logistic regression models for the study data.

Therefore, this study applied the multilevel logistic regression model with region-specific random effects random intercept to account for the intra-class correlation and hence quantify the variation in a card loss that is accounted for by the region variances.

Before fitting the model $M_2$, we used the stepwise variable selection technique to determine the covariates to be included in this model. Then we checked for the presence of multicollinearity among the selected continuous covariates using the variance inflation factors (VIF), excluding those with a VIF greater than 10. In M3, since we have only one independent variable and it is candidate for M4. Then the covariates for the final $M_2$ and M4 models were selected by eliminating those implausible applying the likelihood ratio tests, the Akaike Information Criteria (AIC), and the Bayesian Information Criteria (BIC). Accordingly, of the twenty-four individual covariates considered, only ten variables (educational level of mothers, wealth index, living condition, contraceptive use, ANC use, birth interval, having television, age of the household head, age of the child and place of delivery) were selected for $M_2$, and the region level characteristics (place of residence) and the ten covariates of model $M_2$ and place of residence in $M_3$ were selected for $M_4$.

## Ethical consideration

The study used the data from the 2019 EMDHS obtained from the measure DHS data archive (https://dhsprogram.com/publications/index.cfm) with the appropriate request and permission. The DHS program owns data that are collected following all the necessary ethical

procedures in accordance with the relevant guidelines and regulations. Therefore, for the data used in this analysis all methods were carried out according to relevant guidelines and regulations. The DHS program is authorized to distribute, at no cost, unrestricted survey data files for legitimate academic research. Registration is required for access to data.

## Operational definitions

**Use of vaccination card/mother or child records.** Use of vaccination card/mother or child records means having a child vaccination card or any health records indicating the vaccination status of the child and showing it to the health care provider/any concerned body upon reasonable request.

**Vaccination card loss.** The mother or caregiver is unable to show the child's vaccination card or any relevant record/document, which indicates the vaccination status of the child.

## Results

### Socio-demographic characteristics

This study analyzed Ethiopian mini-DHS data on vaccination card loss among 3208 households in nine regions and two city administrations in Ethiopia. Accordingly, we presented the population characteristics categorized by these regions and city administrations. The finding shows that the largest number of participants were enrolled in Oromia 398 (12.42%), followed by Affar 371 (11.56%), and South Nation Nationality and Population Region(SNNPR) 360 (11.22%), whereas the list from Addis Ababa and Dire Dawa city administrations accounts for 180 (5.64%) and 236 (7.36%), respectively.

The mean (SD) of the age of the respondents shows the highest for the Amhara region 29.48 (6.61), followed by Tigray 28.82 (6.74) and SNNPR 28.63 (6.16) while the lowest is seen for Gambella at 26.57 (6.06) and Affar 26.26 (5.96), with the overall being 27.82 (6.29). The overall average family size is 6.04 (2.41) for all regions and city administrations. More than three-fourths of the respondents, 2442 (76.1%), were rural residents, and the majority, 3025 (94.3%) of the respondents, were in union or married status. Regarding educational status, the finding revealed that more than half (1648) (51.4%) of the respondents have no education. The majority of 2561 (79.8%) of the heads of households were males (**Table 1**).

### Maternal characteristics

The overall mean (SD)for the births in the last five years, the last three years, and the total number of children ever born were 1.65 (0.68), 1.22 (0.42), and 3.73 (2.39), respectively. The finding indicates that the average age at first birth for the respondents is 18.76 (4.12), with the highest in Addis Ababa 22.33 (4.78), Dire Dawa 19.51 (4.13), and the lowest in Gambella 17.57 (3.16) and BenishangulGumuz 17.84 (3.76). The majority, 2892 (90.1%) of the respondents, were not pregnant at the time of data collection. About two-thirds (2134, 66.5%), one-third (1103, 34.4%), and only 379 (13.2%) of the respondents previously used antenatal care (ANC), contraceptives, and postnatal care (PNC), respectively. Concerning place of delivery, closer to half 1493 (46.5%) of the respondents gave birth at home, and more than half 1447 (58%) of the respondents gave birth with a 24–59-month birth interval (**Table 2**).

### Child characteristics

Regarding the child-related variables, the mean (SD)for the number of under-five children in households indicate an overall figure of 1.8 (0.82), with the highest seen in Somalia 2.33 (0.90) and the lowest in the Amhara region 1.42 (0.61). More than two-fifths of 1313 (40.9%) of the

**Table 1. Descriptive statistics for socio-demographic variables of respondents by region in Ethiopia.**

| Variables | Region | | | | | | | | | | | Total |
|---|---|---|---|---|---|---|---|---|---|---|---|---|
| | Tigray | Affar | Amhara | Oromia | Somali | BG | SNNPR | Gambella | Harari | AA | Dire D | |
| N (%) | 261(8.14) | 371 (11.56) | 294(9.16) | 398(12.41) | 321 (10.01) | 289(9.01) | 360(11.22) | 247(7.70) | 251(7.82) | 180(5.64) | 236 (7.36) | 3208 (100) |
| Age (Mean/SD) | 28.82 (6.74) | 26.26 (5.96) | 29.48 (6.61) | 27.66 (6.47) | 28.09 (6.12) | 28.02 (6.82) | 28.63 (6.16) | 26.57(6.06) | 27.39 (5.69) | 28.30 (5.19) | 27.00 (6.1) | 27.82 (6.29) |
| Family size | 5.55(1.94) | 5.99(2.17) | 5.54(2.34) | 6.14(2.35) | 7.09(2.68) | 6.12(2.43) | 6.09(2.12) | 6.36(3.06) | 5.92(2.16) | 5.00(2.03) | 6.06 (2.53) | 6.04 (2.41) |
| Age of the HH head | 39.59 (12.74) | 35.53 (14.35) | 40.13 (13.34) | 37.65 (12.58) | 34.53 (11.77) | 38.58 (13.84) | 38.00 (12.64) | 36.17 (10.36) | 35.28 (11.41) | 35.98 (10.85) | 39.51 (16.9) | 37.34 (13.08 |
| **Place of residence** | | | | | | | | | | | | |
| Urban | 39(14.9) | 64(17.3) | 36(12.2) | 52(13.1) | 42(13.1) | 40(13.8) | 34(9.4) | 54(21.9) | 112(44.6) | 180 (100.0) | 113 (47.9) | 766(23.9) |
| Rural | 222(85.1) | 307(82.7) | 258(87.8) | 346(86.9) | 279(86.9) | 249(86.2) | 326(90.6) | 193(78.1) | 139(55.4) | 0(0.0) | 123 (52.1) | 2442 (76.1) |
| **Educational level** | | | | | | | | | | | | |
| Higher | 14(5.4) | 4(1.1) | 11(3.7) | 5(1.3) | 2(0.6) | 14(4.8) | 11(3.1) | 23(9.3) | 21(8.4) | 49(27.2) | 30(12.7) | 184(5.7) |
| Secondary | 41(15.7) | 13(3.5) | 12(4.1) | 31(7.8) | 7(2.2) | 11(3.8) | 31(8.6) | 49(19.8) | 24(9.6) | 38(21.1) | 39(16.5) | 296(9.2) |
| Primary | 79(30.3) | 84(22.6) | 91(31.0) | 170(42.7) | 48(15.0) | 118(40.8) | 140(38.9) | 111(44.9) | 105(41.8) | 64(35.6) | 70(29.7) | 1080 (33.7) |
| No education | 127(48.7) | 270(72.8) | 180(61.2) | 192(48.2) | 264(82.2) | 146(50.5) | 178(49.4) | 64(25.9) | 101(40.2) | 29(16.1) | 97(41.1) | 1648 (51.4) |
| **Marital status** | | | | | | | | | | | | |
| Single | 17(6.5) | 13(3.5) | 18(6.1) | 20(5.0) | 13(4.0) | 18(6.2) | 11(3.1) | 38(15.4) | 10(4.0) | 10(5.6) | 15(6.4) | 183(5.7) |
| Union/Married | 244(93.5) | 358(96.5) | 276(93.9) | 378(95.0) | 308(96.0) | 271(93.8) | 349(96.9) | 209(84.6) | 241(96.0) | 170(94.4) | 221 (93.6) | 3025 (94.3) |
| **Sex of the HH head** | | | | | | | | | | | | |
| Male | 219(83.9) | 249(67.1) | 265(90.1) | 370(93.0) | 180(56.1) | 251(86.9) | 334(92.8) | 167(67.6) | 197(78.5) | 130(72.2) | 199 (84.3) | 2561 (79.8) |
| Female | 42(16.1) | 122(32.9) | 29(9.9) | 28(7.0) | 141(43.9) | 38(13.1) | 26(7.2) | 80(32.4) | 54(21.5) | 50(27.8) | 37(15.7) | 647(20.2) |
| **Having Television** | | | | | | | | | | | | |
| Yes | 49(18.8) | 33(8.9) | 31(10.5) | 37(9.3) | 12(3.7) | 15(5.2) | 16(4.4) | 20(8.1) | 132(52.6) | 158(87.8) | 108 (45.8) | 611(19.0) |
| No | 212(81.2) | 338(91.1) | 263(89.5) | 361(90.7) | 309(96.3) | 274(94.8) | 344(95.6) | 227(91.9) | 119(47.4) | 22(12.2) | 128 (54.2) | 2597 (81.0) |
| **Having Radio** | | | | | | | | | | | | |
| Yes | 61(23.4) | 65(17.5) | 29(9.9) | 106(26.6) | 27(8.4) | 59(20.4) | 114(31.7) | 34(13.8) | 109(43.4) | 97(53.9) | 72(30.5) | 773(24.1) |
| No | 200(76.6) | 306(82.5) | 265(90.1) | 292(73.4) | 294(91.6) | 230(79.6) | 246(68.3) | 213(86.2) | 142(56.6) | 83(46.1) | 164 (69.5) | 2435 (75.9) |

children were born in the 1–2 birth order. Almost all 3170 (98.8%) of the index children were living with their mothers. Although there are regional variations, the overall sex of the children divides them in half (1582 (49.3%) female and 1626 (50.7%) male). More than one-third of the children were in the age groups of 0–11 months (1107 (34.5%) and 24–35 months (1093 (34.1%)) (**Table 3**).

The data on literacy showed that the highest percentage of study participants in Addis Ababa were able to read and write, whereas the majority of respondents in the Affar region were categorized as not being able to read or write at all (**Fig 1**).

The data on the wealth index combined showed that the highest percentages of study participants in Addis Ababa were categorized in the richest group, whereas the majority of respondents in the Afar region were categorized in the poorest group (**Fig 2**).

**Table 2. Maternal characteristics of the respondents by region and city administrations in Ethiopia.**

| Variables | Region | | | | | | | | | | | Total |
|---|---|---|---|---|---|---|---|---|---|---|---|---|
| | Tigray | Affar | Amhara | Oromia | Somali | BG | SNNPR | Gambella | Harari | AA | Dire D | |
| Birth in the last five years (Mean/SD) | 1.47 (0.53) | 1.93 (0.77) | 1.36(0.53) | 1.67 (0.68) | 2.16(.74) | 1.59(.64) | 1.59(.62) | 1.45(0.59) | 1.69 (0.66) | 1.36 (0.55) | 1.58 (0.62) | 1.65 (0.68) |
| Birth in the last 3 years (Mean/SD) | 1.15 (0.35) | 1.36 (0.52) | 1.07(0.26) | 1.22 (0.42) | 1.42 (0.51) | 1.18 (0.38) | 1.18 (0.38) | 1.09(0.29) | 1.26 (0.45) | 1.11 (0.32) | 1.22 (0.42) | 1.22 (0.42) |
| Total children ever born (Mean/SD) | 3.43 (2.23) | 3.77 (2.35) | 3.49(2.19) | 4.05 (2.70) | 4.98 (2.60) | 3.91 (2.44) | 3.93 (2.25) | 3.36(2.09) | 3.57 (2.29) | 2.17 (1.29) | 3.25 (2.23) | 3.73 (2.39) |
| Age of the mother at first birth (Mean/SD) | 19.36 (3.97) | 18.18 (3.80) | 18.87 (4.43) | 18.15 (3.89) | 18.55 (3.43) | 17.84 (3.76) | 18.86 (4.34) | 17.57 (3.16) | 18.95 (4.29) | 22.33 (4.78) | 19.51 (4.13) | 18.76 (4.12) |
| **Currently pregnant N (%)** | | | | | | | | | | | | |
| No | 242 (92.7) | 322 (86.8) | 285(96.9) | 355(89.2) | 268(83.5) | 270 (93.4) | 327(90.8) | 230(93.1) | 218(86.9) | 164 (91.1) | 211 (89.4) | 2892 (90.1) |
| Yes | 19(7.3) | 49(13.2) | 9(3.1) | 43(10.8) | 53(16.5) | 19(6.6) | 33(9.2) | 17(6.9) | 33(13.1) | 16(8.9) | 25(10.6) | 316(9.9) |
| **ANC use N (%)** | | | | | | | | | | | | |
| Yes | 226 (86.6) | 190 (51.2) | 235(79.9) | 251(63.1) | 70(21.8) | 223 (77.2) | 230(63.9) | 183(74.1) | 178(70.9) | 168 (93.3) | 180 (76.3) | 2134 (66.5) |
| No | 35(13.4) | 181 (48.8) | 59(20.1) | 147(36.9) | 251(78.2) | 66(22.8) | 130(36.1) | 64(25.9) | 73(29.1) | 12(6.7) | 56(23.7) | 1074 (33.5) |
| **Contraceptive use N (%)** | | | | | | | | | | | | |
| Yes | 104 (39.8) | 34(9.2) | 143(48.6) | 157(39.4) | 9(2.8) | 113 (39.1) | 177(49.2) | 73(29.6) | 94(37.5) | 118 (65.6) | 81(34.3) | 1103 (34.4) |
| No | 157 (60.2) | 337 (90.8) | 151(51.4) | 241(60.6) | 312(97.2) | 176 (60.9) | 183(50.8) | 174(70.4) | 157(62.5) | 62(34.4) | 155 (65.7) | 2105 (65.6) |
| **PNC use N (%)** | | | | | | | | | | | | |
| Yes | 48(19.8) | 14(4.5) | 41(14.5) | 42(11.8) | 13(5.1) | 30(11.4) | 49(14.9) | 28(11.9) | 37(16.8) | 35(20.6) | 42(19.9) | 379 (13.2) |
| No | 194 (80.2) | 294 (95.5) | 242(85.5) | 315(88.2) | 244(94.9) | 234 (88.6) | 280(85.1) | 207(88.1) | 183(83.2) | 135 (79.4) | 169 (80.1) | 2497 (86.8) |
| **Place of delivery N (%)** | | | | | | | | | | | | |
| Public health facility | 174 (66.7) | 95(25.3) | 166(56.5) | 158(39.7) | 53(16.5) | 183 (63.3) | 175(48.6) | 117(47.4) | 148(59.0) | 130 (72.2) | 141 (59.7) | 1539 (48.0) |
| Private health facility | 3(1.1) | 5(1.3) | 5(1.7) | 13(3.3) | 11(3.4) | 15(5.2) | 4(1.1) | 28(11.3) | 24(9.6) | 44(24.4) | 24(10.2) | 176(5.5) |
| Home | 84(32.2) | 272 (73.3) | 123(41.8) | 227(57.0) | 257(80.1) | 91(31.5) | 181(50.3) | 102(41.3) | 79(31.5) | 6(3.3) | 71(30.1) | 1493 (46.5) |
| **Birth interval N (%)** | | | | | | | | | | | | |
| > = 60 months | 43(21.8) | 24(8.1) | 84(38.7) | 40(12.9) | 7(2.4) | 49(21.4) | 65(21.5) | 48(26.2) | 29(14.9) | 34(31.8) | 22(13.1) | 445 (17.8) |
| 25–59 months | 131 (66.5) | 163 (55.3) | 106(48.8) | 187(60.1) | 157(54.5) | 145 (63.3) | 181(59.7) | 107(58.5) | 108(55.4) | 50(46.7) | 112 (66.7) | 1447 (58.0) |
| < = 24 months | 23(11.7) | 108 (36.6) | 27(12.4) | 84(27.0) | 124(43.1) | 35(15.3) | 57(18.8) | 28(15.3) | 58(29.7) | 23(21.5) | 34(20.2) | 601 (24.1) |

The result of the study revealed that the level of vaccination card loss in this analysis is 1933 (60.26%) (**Fig 3**).

## Fitted multilevel models

The estimated intercept for model M1 was 0.279 with a confidence interval of (-0.286,.845), while the estimated variance of the region-specific random effects was 0.898 with a confidence interval of (0.377, 2.137). Model 1 (the null model) showed that there was significant variability in the odds of vaccination card loss across regions [$\tau$ = 1.32, p < 0.05]. As indicated in the

**Table 3. Child-related characteristics by region and city administrations in Ethiopia.**

| Variables | Region | | | | | | | | | | | Total |
|---|---|---|---|---|---|---|---|---|---|---|---|---|
| | Tigray | Affar | Amhara | Oromia | Somali | BG | SNNPR | Gambella | Harari | AA | Dire D | |
| Number of U5 (Mean/SD) | 1.65 (0.61) | 2.12 (0.92) | 1.42 (0.61) | 1.82 (0.78) | 2.33 (0.90) | 1.72 (0.73) | 1.79 (0.79) | 1.63(0.83) | 1.81 (0.79) | 1.45 (0.64) | 1.73 (0.74) | 1.80(0.82) |
| **Birth order N (%)** | | | | | | | | | | | | |
| 1–2 | 113(43.3) | 151(40.7) | 123(41.8) | 161(40.5) | 72(22.4) | 111(38.4) | 116(32.2) | 109(44.1) | 111(44.2) | 122(67.8) | 124(52.5) | 1313 (40.9) |
| 3–4 | 80(30.7) | 109(29.4) | 83(28.2) | 89(22.4) | 80(24.9) | 74(25.6) | 118(32.8) | 63(25.5) | 73(29.1) | 49(27.2) | 55(23.3) | 873(27.2) |
| 5+ | 68(26.1) | 111(29.9) | 88(29.9) | 148(37.2) | 169(52.6) | 104(36.0) | 126(35.0) | 75(30.4) | 67(26.7) | 9(5.0) | 57(24.2) | 1022 (31.9) |
| **Living condition N (%)** | | | | | | | | | | | | |
| With mother | 260(99.6) | 363(97.8) | 293(99.7) | 393(98.7) | 314(97.8) | 288(99.7) | 359(99.7) | 242(98.0) | 247(98.4) | 178(98.9) | 233(98.7) | 3170 (98.8) |
| With caretaker | 1(0.4) | 8(2.2) | 1(0.3) | 5(1.3) | 7(2.2) | 1(0.3) | 1(0.3) | 5(2.0) | 4(1.6) | 2(1.1) | 3(1.3) | 38(1.2) |
| **Age of child N (%)** | | | | | | | | | | | | |
| 0–11 months | 90(34.5) | 125(33.7) | 104(35.4) | 131(32.9) | 116(36.1) | 104(36.0) | 125(34.7) | 91(36.8) | 84(33.5) | 55(30.6) | 82(34.7) | 1107 (34.5) |
| 12–23 months | 93(35.6) | 111(29.9) | 99(33.7) | 127(31.9) | 85(26.5) | 83(28.7) | 116(32.2) | 77(31.2) | 73(29.1) | 64(35.6) | 80(33.9) | 1008 (31.4) |
| 24–35 months | 78(29.9) | 135(36.4) | 91(31.0) | 140(35.2) | 120(37.4) | 102(35.3) | 119(33.1) | 79(32.0) | 94(37.5) | 61(33.9) | 74(31.4) | 1093 (34.1) |
| **Sex of the child N (%)** | | | | | | | | | | | | |
| Female | 134(51.3) | 169(45.6) | 143(48.6) | 201(50.5) | 150(46.7) | 142(49.1) | 182(50.6) | 115(46.6) | 147(58.6) | 85(47.2) | 114(48.3) | 1582 (49.3) |
| Male | 127(48.7) | 202(54.4) | 501(51.4) | 197(49.5) | 171(53.3) | 147(50.9) | 178(49.4) | 132(53.4) | 104(41.4) | 95(52.8) | 122(51.7) | 1626 (50.7) |

intraclass correlation coefficient(ICC), 89.7% of the variability in the odds of a child having a vaccination card lost is due to region-level factors. The variation in vaccination card loss in Model 2 remained significant [$\tau = 0.171$, p < 0.001], with 31.5% of the variance among observations being attributed due to household. The values of these criteria suggest that $M2$ is a preferred model to fit the study. Therefore, in what follows, we only report results based on $M2$. The fit model selection criteria for fitted $M1$, $M2$, $M3$, and $M4$ are displayed in **Table 4**.

## Factors associated with vaccination card loss

The odds ratio with 95% confidence intervals (CI) in brackets for model $M2$ is given in **Table 5**. The p-values show that the ten individual characteristics, namely the educational level of mothers, wealth index, living condition, contraceptive use, ANC use, birth interval, having television, age of the household head, age of the child, and place of delivery, were significantly associated with the odds of vaccination card loss among the study population. Furthermore, from the region-level characteristics, residence was also statistically significant (p-value < 0.05).

After adjusting for other variables, the odds of vaccination card loss are 2.365 times higher in mothers who attended secondary education (95% CI: 1.289, 4.336), 2.166 times higher in those who attended primary education (95% CI: 1.268, 3.700), and 2.239 times higher in those with no education (95% CI: 1.297, 3.864) when compared with those who attended higher education. Regarding the wealth index, the odds of vaccination card loss is 2.089 times higher for respondents in the poorest wealth index category than those in the highest category (95% CI: 1.432, 3.048).

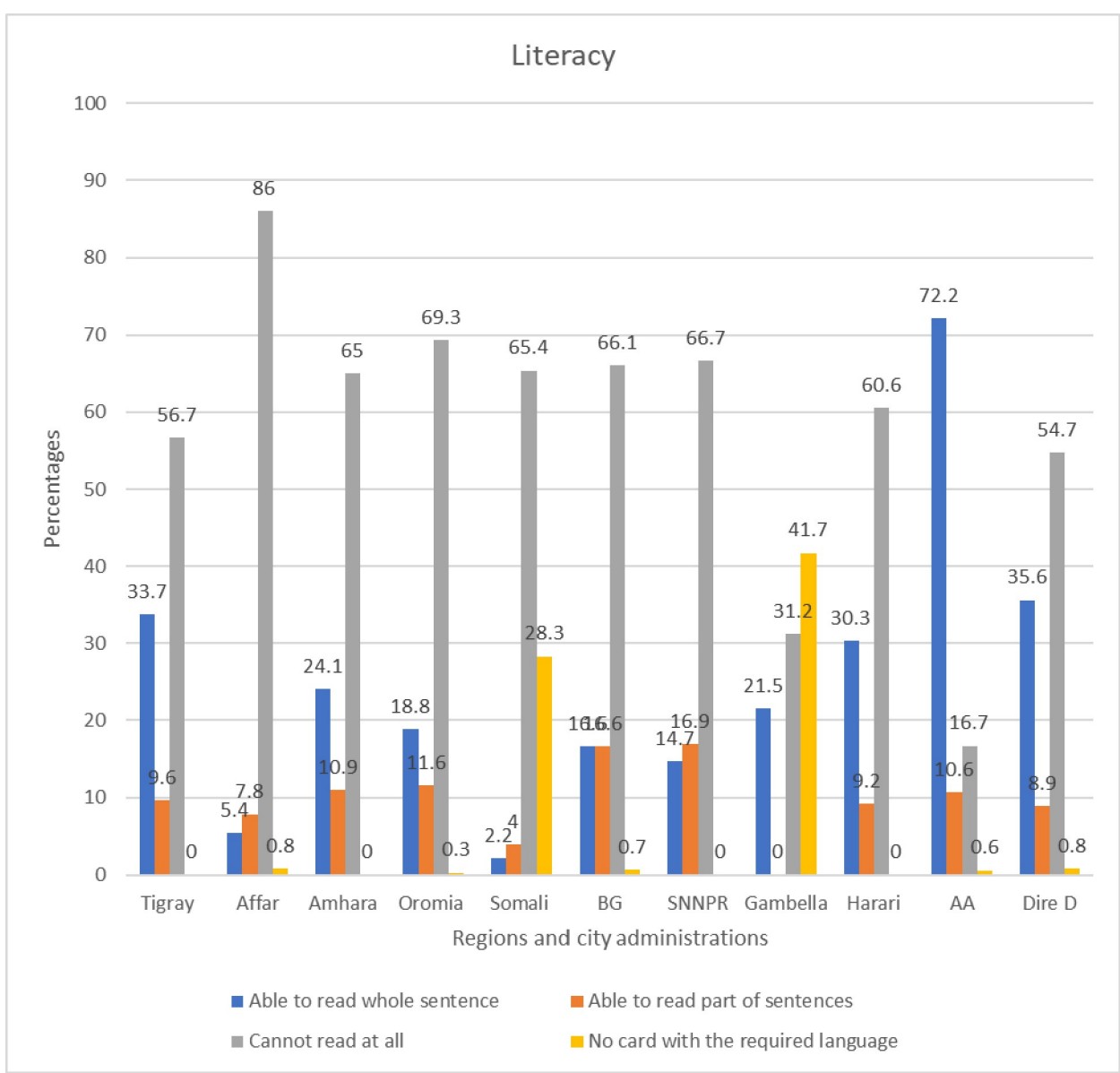

**Fig 1. Literacy status of respondents by region and city administrations in Ethiopia.**

Concerning the age of the respondent, as the age increases by one unit, the odds of vaccination card loss decrease by 0.985 units (95% CI: 0.977, 0.993). Regarding ANC use, the odds of vaccination card loss in ANC non-users are 2.047 times higher when compared with ANC users (95% CI: 1.605, 2.609). The analysis also revealed that the odds of vaccination card loss is 6.749 times higher for children living with their caretakers compared with those living with their mothers (95% CI: 1.425, 13.654). As to the source of information, the odds of vaccination card loss is 1.384 times higher for those respondents who did not have a television when compared with those who have a television set at home (95% CI: 1.150, 1.664). Regarding the birth interval, the odds of vaccination card loss for respondents who gave the next birth after 25–59 months is 1.325 times higher compared to those who gave the next birth after 60 months (with a 95% CI of 1.027, 1.710). It was revealed that the odds of vaccination card loss is 1.696 times

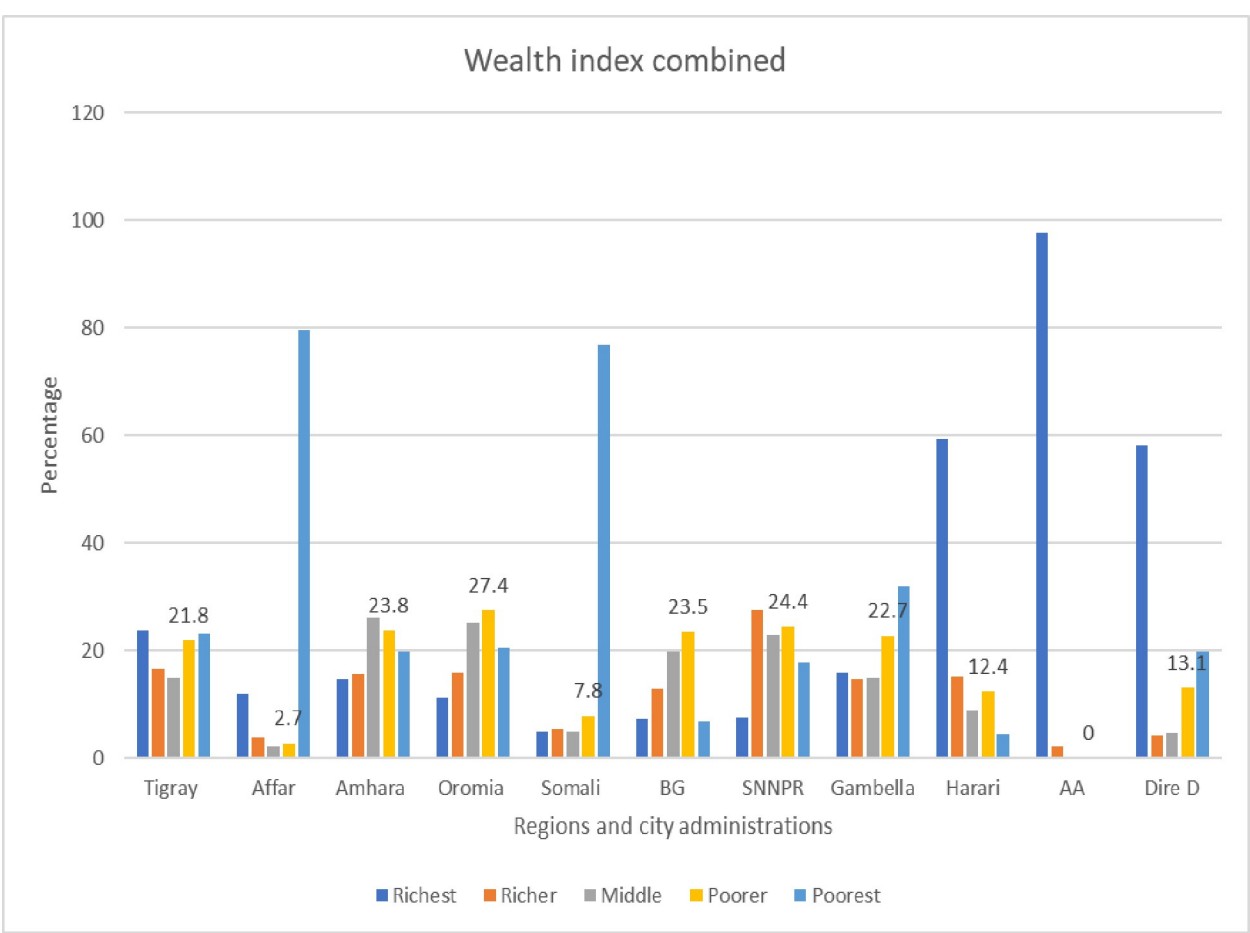

**Fig 2. Wealth index combined of respondents by region and city administrations in Ethiopia.**

higher for respondents who deliver at private health facilities (with a 95% CI of 1.086, 2.648) and 1.985 times higher for those who deliver at home (with a 95% CI of 1.579, 2.497) when compared to those delivered at the public health facility. The analysis also implies that the odds of vaccination card loss among contraceptive non-users is 1.295 times higher than those of contraceptive users (with a 95% CI of 1.042, 1.609). Finally, in relation to the age of children, the odds of vaccination card loss among children aged 12–23 months is 1.577 times higher (with a 95% CI of 1.252, 1.985) and 2.282 times higher for those aged 24–35 months (with a 95% CI of 1.803, 2.889) compared with those aged 0–11 months, respectively (**Table 5**).

## Discussion

Vaccination card loss is one of the critical challenges in the health system to ensure vaccine effectiveness and coverage. This study assessed the level of vaccination card loss and associated factors among mothers of 0–35-month-old children in Ethiopia based on the data from mini-EDHS 2019. In this study prevalence/magnitude of vaccination card loss in mini EDHS-2019 was found to be 60.26% which is similar to the study conducted in Pakistan (67%) [14]. Ethiopia EDHS 2005(63%), EDHS 2011(71%) [21]. Higher than study conducted in Ghana (10.5%) [22], Nepal (17.8%) [23], Senegal (26.3%) [5], Cameron (33% [24]), Pakistan (33.8) [1], Uganda (34%) [25] and Nigeria (36.7%) [26]. This high vaccination card loss of 62% among

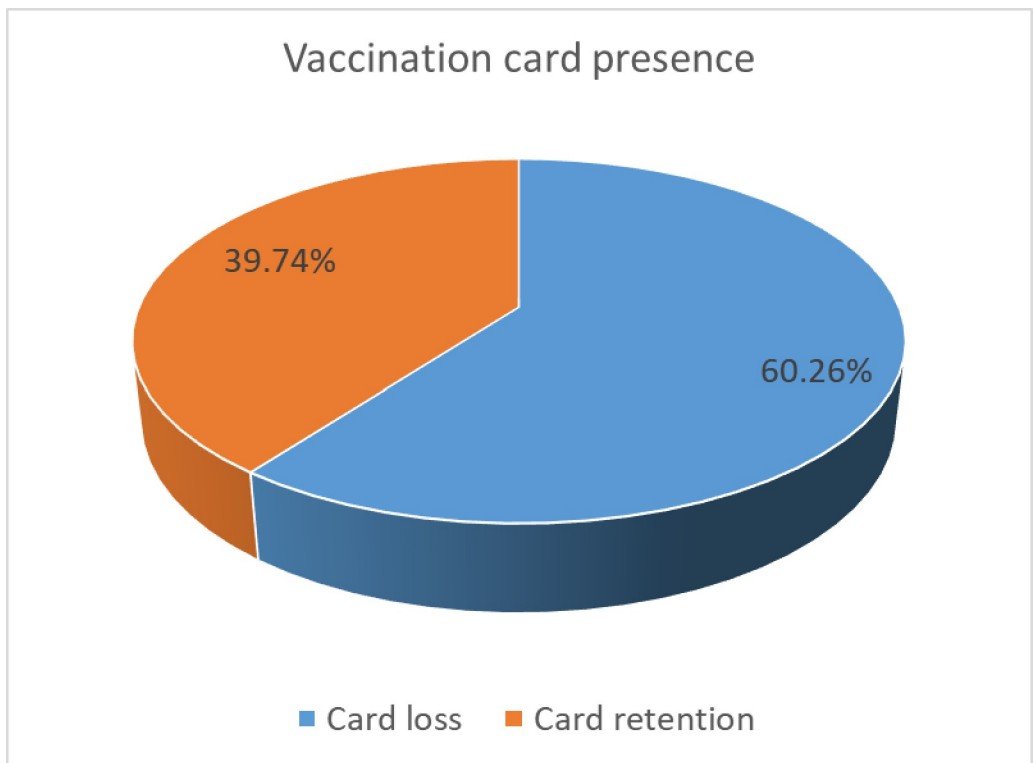

**Fig 3. Presence of vaccination cards at households in Ethiopia.**

children less than 35 months indicates that even if vaccination was introduced in 1980 in the Ethiopian health system to prevent vaccine-preventable disease and different efforts were made to tackle this public health problem, the health system demands extra commitment and proven strategies to put in place these cost-effective interventions. Still in Ethiopia, there are significant numbers of children under three years who are not sure which vaccine and doze they receive due to vaccination card loss; the health care provider, the parents, and the whole community are at risk for epidemics like measles due to loss of vaccination card to trace defaulters.

In this study, respondents' with low educational status are more likely to experience child vaccination card loss, households who had attended no education, primary education, and secondary education are more likely to lose vaccination cards compared with respondents who had attended higher education. This finding is supported by research from India [13] and Nepal [2] in which mothers with no education and mothers who attended primary and

**Table 4. Fit statistics for models with cluster-specific random effects *bj* only (*M*1), with the individual / household level predictors and *bj* (*M*2), and *M*2 with cluster-specific predictors (*M*4).**

| Fit statistics | Fitted models | | | |
|---|---|---|---|---|
| | M1 | M2 | M3 | M4 |
| Region level variance (95%CI) | 0.898(.377,2.137) | .315(.121,.827) | .593(.246,1.430) | .312(.119,.816) |
| Median OR (95%CI) | 1.322(.751,2.328) | .171(.084 .349) | .235(.132 .417) | .142(.062, .325) |
| -2log(Lik) | -1942.629 | -1313.921 | -1893.526 | -1313.552 |
| AIC (smaller is better) | 3889.258 | 2667.842 | 3793.053 | 2669.103 |
| BIC (Smaller is better) | 3901.405 | 2784.267 | 3811.273 | 2791.349 |

**Table 5. Estimated odds ratios (95% confidence intervals) for fitted models about the predictors of vaccination card loss among children 0–35 months in Ethiopia.**

| Variable | Model I | Model II | Model III | Model IV |
|---|---|---|---|---|
| **Fixed effect** | **IOR (95%CI)** | **IOR (95%CI)** | **IOR (95%CI)** | **IOR (95%CI)** |
| **Types of place of residence** | | | | |
| Urban | | | 1 | 1 |
| Rural | | | 2.747 (2.248,3.357) * | 1.156(.831,1.609) |
| **The individual Level Variables** | | | | |
| **The educational level of the mother** | | | | |
| Higher | | 1 | | 1 |
| Secondary | | 2.365(1.289,4.336)* | | 2.367(1.290,4.341)* |
| Primary | | 2.166(1.268, 3.700)* | | 2.163(1.266,3.695)* |
| Uneducated | | 2.239(1.297, 3.864)* | | 2.238(1.297, 3.863)* |
| **Wealth index** | | | | |
| Richest | | 1 | . | 1 |
| Richer | | 1.044(.716,1.521) | | 0.963 (0.633, 1.464) |
| Middle | | 1.241(.843,1.827) | | 1.126 (0.722, 1.757) |
| Poorer | | 1.439(.987,2.098) | | 1.303 (0.840, 2.023) |
| Poorest | | 2.089(1.432,3.048)* | | 1.886 (1.211, 2.939)* |
| **Age of respondent** | | 0.985(0.977, 0.993)* | | 0.985 (0.978, 0.993)* |
| **ANC Use** | | | | |
| Yes | | 1 | | 1 |
| No | | 2.047(1.605,2.609)* | | 2.044 (1.603, 2.606)* |
| **Living condition** | | | | |
| With mother | | 1 | | 1 |
| With caretaker | | 6.749(1.425,13.654)* | | 6.504 (1.528, 14.231)* |
| **Having Television** | | | | |
| Yes | | 1 | | 1 |
| No | | 1.384(1.150,1.664)* | | 1.387 (1.153, 1.667)* |
| **Birth interval** | | | | |
| > = 60 months | | 1 | | 1 |
| 25–59 months | | 1.325(1.027,1.710)* | | 1.325 (1.027, 1.710)* |
| < = 24 months | | 1.289(.946,1.757) | | 1.290 (0.947, 1.759) |
| **Place delivery** | | | | |
| Public HF | | 1 | | 1 |
| Private HF | | 1.696(1.086,2.648)* | | 1.716 (1.098, 2.681)* |
| Home | | 1.985(1.579, 2.497)* | | 1.972 (1.567, 2.481)* |
| **Contraceptive use** | | | | |
| Yes | | 1 | | 1 |
| No | | 1.295 (1.042,1.609)* | | 1.296 (1.043, 1.610)* |
| **Age of child** | | | | |
| 0–11 months | | 1 | | 1 |
| 12–23 months | | 1.577(1.252, 1.985)* | | 1.577(1.253, 1.986)* |
| 24–35 months | | 2.282(1.803, 2.889)* | | 2.281(1.802, 2.888)* |

secondary education have not retained the vaccination card of their children; they don't consider the vaccination card a valuable document throughout the life of their child and even if it contains essential information about vaccination status and the next appointment schedule as compared with mothers with a higher educational level.

In this study, respondents with the poorest wealth index are more likely to experience the loss of their child's vaccination card as compared with respondents in the highest wealth

index. This result was supported by findings from Nepal [2] and Cameron [24]. In this study, mothers from the poorest wealth index category reported higher vaccination card losses as compared to mothers from the highest wealth index. Mothers from the poorest wealth index category have low economic status. Due to their low economic status, they focus on their daily earnings and assurance of basic expenses, other than focusing on the importance of vaccination and keeping and securing vaccination cards. The housing condition of mothers in the poorest wealth index in our context is not safe to keep and protect the vaccination card. The housing is exposed to sun, rain, and rats, which affect the quality and safety of the vaccination card.

The age of the respondents was another determinant factor for the vaccination card loss. In this study as the age of respondents increases card loss decreases. As the age of respondents increases, there is the possibility of repeated experience on the importance of having vaccination cards of their children for different events, like as prerequisite evidence for joining the schools and for certificates with regard to vital events in the country. These all experiences enhance vaccination card retention as age increases. A study conducted in Cameron [24] contradicts this finding, which shows that as respondents age increases, vaccination card loss increases.

In this study, mothers from rural residences had a positive association with vaccination card loss. Mothers from rural residences were more likely to lose vaccination cards as compared with mothers from urban areas. The finding of this study is contrary to the study done in Nepal [2] where there is higher vaccination card retention among rural residents. Mothers from rural areas have no suitable places to keep their child's vaccination cards as compared with mothers from urban residences. Mothers from rural residences, due to their low socioeconomic level, poor housing condition, and low education level, give little attention to the safekeeping of vaccination cards.

In this study, the age of children had an association with vaccination card loss, children aged 12–23 months and children aged 24–35 were more likely to lose vaccination cards as compared with children aged less than 12 months. This finding is supported by findings from Bangladesh [27], Pakistan [1], Senegal [5], Ghana [22] and Cameron [24]. While the age of a child increases, vaccination card loss increases. Mothers at an early stage have concerns about vaccination cards.

In this study, children living with their caretakers were more likely to lose vaccination cards as compared with those living with their mothers. This finding is supported by the study conducted in Cameron children living with biological parents were more likely to retain vaccination cards compared to children living with caretakers [22]. Caretakers might have low contact and follow-up to trace and check the vaccination card due to being busy with other extra tasks.

In this study, mothers who did not attend ANC reported higher vaccination card losses as compared with mothers who attended ANC follow-up. The finding of this study is supported with studies conducted in Nepal [23] and Senegal [5]. Mothers who do not attend ANC have no room to contact and communicate with healthcare providers about the importance of health services provided during and immediately after birth for their infants. ANC follow-up creates a good client-provider interaction to discuss issues related to overall child health and the importance of vaccination cards during child health follow-up. ANC packages focus on enhancing the health of the mother and the child. One approach to enhancing the health of the child is providing childhood vaccination. To access full vaccination, doze vaccination cards are used as evidence, and healthcare providers communicate this with mothers during ANC follow-ups. Mothers who don't visit health facilities during pregnancy lack all the above-listed benefits.

In this study, place of delivery had an association with vaccination card loss. Mothers who had delivered in home and private health facilities were more likely to lose vaccination cards as compared with mothers who delivered in public health facilities. Similar findings were reported in a study conducted in Nepal [23], India [13], and Senegal [5] in which vaccination card losses were higher among children who were born in the home and private health facilities compared with children born in the public health facilities. Children who are born in public health facilities have the possibility of being introduced to BCG and poilio 0 at birth as packages during the first six hours of PNC follow-up at public health facilities, but this is not possible in cases of private and home delivery where there is no vaccine supply. Late introduction of vaccines after birth creates low demand for vaccination among mothers, which in turn creates a low bound between the mothers and health facilities to deal with and discuss the importance of vaccination cards to follow the vaccination status of their child.

In this study, mothers without television had a positive association with vaccination card loss. Mothers who do not have media access, like television, have higher vaccination card losses as compared with mothers who have access to television. Messages related to the importance of vaccination cards and how to keep valuable documents are delivered through television, and mothers who do not have access to such important information to keep vaccination cards were more likely to loss vaccination card.

In this study, recording birth interval, mothers who gave next delivery before 59 months were more likely to loss vaccination card as compared with mothers who delivered the next baby after 60 months. Having an additional child in a short birth interval and accompanying the other child make the mother busy and forget about safekeeping of the vaccination card.

In this study, contraceptive non-user mothers had more likely to loss vaccination card as compared with mother who uses contraceptive. Mothers who use contraceptives have experienced the importance of cards during family planning appointments. This lived experience of mothers during family planning appointments on the use of cards enhances their ability to keep their children's vaccination cards to use during immunization schedule appointments. The limitation of this study is investigating the regional-level variation without considering the district-level.

## Conclusion and recommendation

Vaccination cards are essential tools for tracking and addressing effective immunization programs. However, losing the vaccination card is a significant challenge to the effective immunization of children 0–35 months of age in Ethiopia.

In this study, about 60.26% of vaccination cards were lost among mothers of 0–35-month-old children in Ethiopia. Based on our findings, living in a rural residence, educational level of the mother, age 24–35 months, wealth index, age of household head, ANC use, living condition (living with a caretaker), birth interval, place of delivery, contraceptive use, and age of the child are significantly associated with vaccination card retention among mothers of 0–35 months age. According to this finding, vaccination card loss by mothers of 0–35-year-old children in Ethiopia is higher than studies conducted in other regions.

Based on our findings, we recommend to the FMoH that campaign strategies be redesigned to improve the knowledge and awareness of mothers and/or caretakers about the implication of vaccination card loss on an effective vaccination program.

FMoH gives due emphasis and designs intervention mechanisms for mothers who reside in rural, uneducated, non-contraceptive user, have non-ANC follow-up, and are delivered outside public health facilities to reduce vaccination card loss.

Mothers of children 0–35 months old must keep their children's vaccination cards in a safe and easily accessible location. It is also important to update the vaccination card with any new vaccinations received and to ensure that it is brought to all healthcare appointments.

Healthcare providers should be encouraged to provide children's vaccination cards to their mother or caretaker with reminders about upcoming vaccinations and to assist children in obtaining a replacement vaccination card if the original is lost or damaged.

In addition, our recommendation to the researchers is to investigate the cause of vaccination card loss differences between rural and urban, the poorest and richest families; children live with a mother and caretaker; a mother delivers at home; and health facilities.

## Acknowledgments

We would like to thank Jimma University for providing the chance to conduct a study on this topic. We also thank the EPHI, the ICF, and the global DHS database for the data they collected and provided online in collaboration with the CSA and the FMoH.

## Author Contributions

**Conceptualization:** Zerihun Kura Edossa, Belay Erchafo Lubago, Minale Fekadie Baye, Rediet Kidane Alemu, Abebe Abera Tesema, Fira Abamecha, Yibeltal Siraneh, Dessalegn Tamiru, Negalign Berhanu Bayou, Gurmesa Tura Debelew.

**Data curation:** Zerihun Kura Edossa, Belay Erchafo Lubago, Minale Fekadie Baye, Rediet Kidane Alemu, Abebe Abera Tesema.

**Formal analysis:** Zerihun Kura Edossa, Belay Erchafo Lubago, Minale Fekadie Baye, Rediet Kidane Alemu, Abebe Abera Tesema.

**Investigation:** Zerihun Kura Edossa, Belay Erchafo Lubago, Minale Fekadie Baye, Rediet Kidane Alemu, Abebe Abera Tesema.

**Methodology:** Zerihun Kura Edossa, Belay Erchafo Lubago, Minale Fekadie Baye, Rediet Kidane Alemu, Abebe Abera Tesema, Fira Abamecha, Yibeltal Siraneh, Dessalegn Tamiru, Negalign Berhanu Bayou, Gurmesa Tura Debelew.

**Software:** Zerihun Kura Edossa, Belay Erchafo Lubago, Minale Fekadie Baye, Rediet Kidane Alemu, Abebe Abera Tesema.

**Writing – original draft:** Zerihun Kura Edossa, Belay Erchafo Lubago, Minale Fekadie Baye, Rediet Kidane Alemu, Abebe Abera Tesema, Fira Abamecha, Yibeltal Siraneh, Dessalegn Tamiru, Negalign Berhanu Bayou, Gurmesa Tura Debelew.

**Writing – review & editing:** Zerihun Kura Edossa, Belay Erchafo Lubago, Minale Fekadie Baye, Rediet Kidane Alemu, Abebe Abera Tesema, Fira Abamecha, Yibeltal Siraneh, Dessalegn Tamiru, Negalign Berhanu Bayou, Gurmesa Tura Debelew.

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
