## [Decision Letter · Decision Letter 0]

7 Aug 2024

PONE-D-24-15363Vaccination Card Loss and Associated Factors in Ethiopia: A Multilevel Analysis using Ethiopia Mini Demographic and Health Survey 2019 DataPLOS ONE

Dear Dr. Erchafo,

Thank you for submitting your manuscript to PLOS ONE. After careful consideration, we feel that it has merit but does not fully meet PLOS ONE’s publication criteria as it currently stands. Therefore, we invite you to submit a revised version of the manuscript that addresses the points raised during the review process. Please review and address the reviewer's comment .

We look forward to receiving your revised manuscript.

Kind regards,

Dipendra Khatiwada, MD

Academic Editor

PLOS ONE

Journal Requirements:

Additional Editor Comments:

Please address the reviewer's comment

Reviewers' comments:

Reviewer's Responses to Questions

**Comments to the Author**

1. Is the manuscript technically sound, and do the data support the conclusions?

Reviewer #1: Yes

Reviewer #2: Yes

2. Has the statistical analysis been performed appropriately and rigorously? 

Reviewer #1: Yes

Reviewer #2: Yes

3. Have the authors made all data underlying the findings in their manuscript fully available?

Reviewer #1: Yes

Reviewer #2: Yes

4. Is the manuscript presented in an intelligible fashion and written in standard English?

Reviewer #1: Yes

Reviewer #2: Yes

5. Review Comments to the Author

Reviewer #1: Vaccination card retention is a very interesting and under-looked researched topic. The researchers have tried to bring about factors associated with it.

My comments are:

Ln 52 - 54: (Recommendation in abstract): HCP should be encouraged to provide vaccination cards to mothers or caregivers of children.

-> This is not based on the findings of the study.

Statistical Analysis:

Ln 164: .. applied the multilevel logistic regression model with region-specific random effects..

-> Need to further clarify if it is - fixed effect random intercept, or random effect random intercept model.

Ln 167 - 177: Details of model M3 is required. It is not clear how the model was formed.

Result

Ln 204, Ln 230, Ln 262: The mean plus standard deviation ...

-> The SD has been kept in parenthesis rather than after plus sign. So, mention accordingly.

Ln 258, Ln 292: Table 2,3: Do mention the notation Mean(SD) or N(%) in the table as well.

Figure 3: caption: presence of vaccination ...

-> Keep in sentence case (Presence of ..)

Ln 333: ... with 31.5% of the variance among observations being attributed

-> Complete the sentence.

Ln 337: Table 4 caption does not mention about M3 model.

Discussion:

Ln 412: ..education has a positive association with vaccination card loss.

-> the association can be better called negative, as with increase education, there is reduced vaccination card loss.

Ln 421: ... wealth index had a positive association..

-> we can call it positive association with vaccination card retention rather than the card loss.

Ln 432: The age of respondents was another determinant factor for vaccination card loss.

-> The table 5 reports age of HH, while age of respondent is mentioned here.

-> And, the contrasting findings should be discussed, with plausible reasons for the finding of decrease in vaccination card loss with increasing age- reported in this study.

Ln 433-434: A study in Cameron. Contrasts this findings...

-> Remove the period and correct the sentence.

Ln 437-438: the findings of this study is contrary to the study done in Nepal Nepal where higher vaccination card retention ....

-> Reframe and correct the sentence grammatically.

Ln 481- 483: .. recording birth interval, mothers who gave ..

-> This finding also requires discussion.

Limitation - seems to be missing (if there is any in the study). The limitations can be regarding the model selection or others if any.

Reviewer #2: The study's sampling design and methods could have been more detailed to eliminate any biases.

Details on how the random effects were specified and the rationale for choosing specific levels would enhance transparency.

The methodology could give a detailed explanation of how potential confounders were controlled and whether there was multicollinearity among variables.

6. PLOS authors have the option to publish the peer review history of their article (what does this mean?). If published, this will include your full peer review and any attached files.

Reviewer #1: No

Reviewer #2: No

---

## [Author Response · Author response to Decision Letter 0]

6 Sep 2024

PONE-D-24-15363

Vaccination Card Loss and Associated Factors in Ethiopia: A Multilevel Analysis using Ethiopia Mini Demographic and Health Survey 2019 Data

PLOS ONE

Dear Dr. Erchafo,

Thank you for submitting your manuscript to PLOS ONE. After careful consideration, we feel that it has merit but does not fully meet PLOS ONE’s publication criteria as it currently stands. Therefore, we invite you to submit a revised version of the manuscript that addresses the points raised during the review process. 

Please review and address the reviewer's comment .

We look forward to receiving your revised manuscript.

Kind regards,

Dipendra Khatiwada, MD

Academic Editor

PLOS ONE

Dear Academic Editor and Reviewers:

Thank you for providing us an opportunity to revise and resubmit our paper. Below we address all the questions and concerns raised by the academic editor and reviewers in a point-by-point fashion.

Journal Requirements:

 Response: the revised manuscript follows the PLOS ONE format.

 Response: The original data set of 2019 EMDHS is available from EDHs program on request using the following Link (https://dhsprogram.com/publications/index.cfm)

 Response: we appreciate the comment and we have incorporated the updated references in the revised copy.

Reviewer #1: Vaccination card retention is a very interesting and under-looked researched topic. The researchers have tried to bring about factors associated with it.

My comments are:

Ln 52 - 54: (Recommendation in abstract): HCP should be encouraged to provide vaccination cards to mothers or caregivers of children.

-> This is not based on the findings of the study.

 Response: we appreciate the comment and we have incorporated it in the revised copy.

StatisticalAnalysis:

Ln 164: .. applied the multilevel logistic regression model with region-specific random effects..

-> Need to further clarify if it is - fixed effect random intercept, or random effect random intercept model.

 Response: random effect random intercept model.

Ln 167 - 177: Details of model M3 is required. It is not clear how the model was formed.

 Response: In M3, since we have only one independent variable at regional level (Residence), we didn’t apply variable selection technique to determine the covariates to be included in this model. Because of that we didn’t discuss about m3. However, it is candidate for M4. So, now we have incorporated it in the revised copy.

Result

Ln 204, Ln 230, Ln 262: The mean plus standard deviation ...

-> The SD has been kept in parenthesis rather than after plus sign. So, mention accordingly.

 Response: we appreciate the comment and we have incorporated it in the revised copy.

Ln 258, Ln 292: Table 2,3: Do mention the notation Mean(SD) or N(%) in the table as well.

 Response: we appreciate the comment and we have incorporated it in the revised copy.

Figure 3: caption: presence of vaccination ...

-> Keep in sentence case (Presence of ..)

 Response: we appreciate the comment and we have incorporated it in the revised copy.

Ln 333: ... with 31.5% of the variance among observations being attributed

-> Complete the sentence.

 Response: we appreciate the comment and we have incorporated it in the revised copy.

Ln 337: Table 4 caption does not mention about M3 model.

 Response: As we have mentioned above, M3 was applied at reginal level and had only one independent variable. So, M3 was not candidate for model selection.

Discussion:

Ln 412: ..education has a positive association with vaccination card loss.

-> the association can be better called negative, as with increase education, there is reduced vaccination card loss.

Response: we have addressed it in the revised copy as follows:

In this study, respondents’ with low education status are more likely to experience child vaccination card loss.

Ln 421: ... wealth index had a positive association..

-> we can call it positive association with vaccination card retention rather than the card loss.

Response: we have addressed it in the revised copy as follows:

In this study, respondents with the poorest wealth index status are more likely to experience the loss of their child's vaccination card.

Ln 432: The age of respondents was another determinant factor for vaccination card loss.

-> The table 5 reports age of HH, while age of respondent is mentioned here.

-> And, the contrasting findings should be discussed, with plausible reasons for the finding of decrease in vaccination card loss with increasing age- reported in this study.

Response: we have addressed it in the revised copy as follows:

As the age of respondents increases, there is the possibility of repeated experiences on the importance of having vaccination cards of their children for different events, like as prerequisite evidence for joining the schools and for certificates with regard to vital events in the country. These all experiences enhance vaccination card retention as age increases. 

Ln 433-434: A study in Cameron. Contrasts this finding...

-> Remove the period and correct the sentence.

 Response: we appreciate the comment and we have incorporated it in the revised copy.

Ln 437-438: the findings of this study is contrary to the study done in Nepal Nepal where higher vaccination card retention ....

-> Reframe and correct the sentence grammatically.

 Response: we appreciate the comment and we have incorporated it in the revised copy.

Ln 481- 483: .. recording birth interval, mothers who gave ..

-> This finding also requires discussion.

Response: we have addressed it in the revised copy as follows:

Having an additional child in a short birth interval and accompanying the other child make the mother busy and forget about safekeeping of the vaccination card.

Limitation - seems to be missing (if there is any in the study). The limitations can be regarding the model selection or others if any.

Response: we have addressed it in the revised copy as follows:

The limitation of this study is investigating the regional-level variation without considering the district-level. 

Reviewer #2: The study's sampling design and methods could have been more detailed to eliminate any biases.

 Response: we appreciate the comment, and we used the EDHS sampling design and methods, and the data source was also EDHS.

Details on how the random effects were specified and the rationale for choosing specific levels would enhance transparency.

 Response: we appreciate the comment and we have incorporated it.

The methodology could give a detailed explanation of how potential confounders were controlled and whether there was multicollinearity among variables.

 Response: we appreciate the comment and we have incorporated it.

---

## [Editor Report · Decision Letter 1]

27 Sep 2024

Vaccination Card Loss and Associated Factors in Ethiopia: A Multilevel Analysis using Ethiopia Mini Demographic and Health Survey 2019 Data

PONE-D-24-15363R1

Dear Dr. Erchafo,

We’re pleased to inform you that your manuscript has been judged scientifically suitable for publication and will be formally accepted for publication once it meets all outstanding technical requirements.

Kind regards,

Dipendra Khatiwada, MD

Academic Editor

PLOS ONE
---

## [Editor Report · Acceptance letter]

20 Nov 2024

PONE-D-24-15363R1 

PLOS ONE

Dear Dr. Erchafo, 

I'm pleased to inform you that your manuscript has been deemed suitable for publication in PLOS ONE. Congratulations! Your manuscript is now being handed over to our production team.

Kind regards, 

on behalf of

Dr. Dipendra Khatiwada 

Academic Editor

PLOS ONE